# Evaluation of the Biological Effect of Non-UV-Activated Bergapten on Selected Human Tumor Cells and the Insight into the Molecular Mechanism of Its Action

**DOI:** 10.3390/ijms242115555

**Published:** 2023-10-25

**Authors:** Magdalena Bartnik, Adrianna Sławińska-Brych, Magdalena Mizerska-Kowalska, Barbara Zdzisińska

**Affiliations:** 1Department of Pharmacognosy with Medicinal Plants Garden, Medical University of Lublin, Chodźki 1 Street, 20-093 Lublin, Poland; 2Department of Cell Biology, Institute of Biological Sciences, Maria Curie-Skłodowska University, Akademicka 19 Street, 20-033 Lublin, Poland; adrianna.slawinska-brych@mail.umcs.pl; 3Department of Virology and Immunology, Institute of Biological Sciences, Maria Curie-Skłodowska University, Akademicka 19 Street, 20-033 Lublin, Poland; magdalena.mizerska-kowalska@mail.umcs.pl (M.M.-K.); barbara.zdzisinska@mail.umcs.pl (B.Z.)

**Keywords:** bergapten, apoptosis, osteosarcoma (Saos-2 and HOS), AKT, Bax/Bcl-2/caspases pathway, cell cycle block in G2 phase, *Peucedanum tauricum*, multiple myeloma (RPMI8226 and U266), colorectal adenocarcinoma (HT-29 and SW620)

## Abstract

There is some evidence that non-photoactivated psoralens may be active against breast and colon tumor cells. Therefore, we evaluated the antiproliferative, proapoptotic, and anti-migrative effect of 5-methoxypsoralen (5-MOP) isolated from *Peucedanum tauricum* MB fruits in human colorectal adenocarcinoma (HT-29 and SW620), osteosarcoma (Saos-2 and HOS), and multiple myeloma (RPMI8226 and U266). Dose- and cell-line-dependent effects of 5-MOP on viability and proliferation were observed, with the strongest inhibitory effect against Saos-2 and a moderate effect against the HOS, HT-29, and SW620 cells. Multiple myeloma showed low sensitivity. The high viability of human normal cell cultures (HSF and hFOB) in a wide range of 5-MOP concentrations tested (6.25–100 µM) was confirmed. Moreover, the migration of treated Saos-2, SW620, and HT-29 cell lines was impaired, as indicated via a wound healing assay. Flow cytometry analysis conducted on Saos-2 cells revealed the ability of 5-MOP to block the cell cycle in the G2 phase and trigger apoptosis, which was accompanied by a loss of mitochondrial membrane potential, caspases (-9 and -3) activation, the altered expression of the Bax and Bcl-2 proteins, and decreased AKT phosphorylation. This is the first report evaluating the antiproliferative and antimigratory impact of non-UV-activated bergapten on the abovementioned (except for HT-29) tumor cells, which provides new data on the potential role of 5-MOP in inhibiting the growth of various types of therapeutic-resistant cancers.

## 1. Introduction

Cancer is a serious health problem with a high mortality rate worldwide and is extremely common. Some types of cancer are especially chemo-resistant, metastatic, difficult to cure, and even still remain incurable, such as colorectal cancers, osteosarcomas, and multiple myelomas [1]. Besides genetic factors, the epigenetic influence of a polluted environment, stress in daily life, and poor dietary habits remain the key risk factors of the constantly increasing cancer incidence. The occurrence of colorectal or colon cancer has been progressively increasing in recent years, and it has become the third most common type of cancer worldwide and a leading cause of cancer-related death [2].

Osteosarcoma is the most common primary bone tumor in children and adolescents. The main factors for the poor prognosis of osteosarcoma are recurrences and metastases (resulting in a 5-year survival rate in patients of less than 20%) [3]. Unfortunately, this sad picture is frequently due to the failure of chemotherapy associated with chemo-resistance phenomena [4].

Multiple myeloma is the second most common hematological cancer, and it still remains incurable. For the treatment of this clinically heterogeneous plasma cell cancer (mostly restricted to the bone marrow), up to the end of the year 2020, 92 compounds with effects on at least one human myeloma cell line were found [1]. This type of cancer is highly metastatic and aggressive; therefore, multitarget therapies, not restricted to one compound/therapeutic methodology, are required, and new therapeutic substances are still needed.

One of the important ways to fight cancer (an alternative to surgery or support to chemo- and radiotherapy) is the use of natural products, which have constantly attracted attention due to their large number of chemical structures, multiple biological activities, fewer side effects, and high safety aspects compared to synthetic drugs [1,5,6]. Compounds derived from natural sources such as plants, animals, or microorganisms have always been some of the most important and relevant assets in drug discovery [7]. Used by humans since ancient times, these natural compounds have the power to maintain and improve health and prevent diseases [8]. Many experiments have highlighted the protective function of bioactive compounds of plant origin. When combined with other therapeutic agents, plant-derived metabolites can act synergistically and reduce the side effects of the chemotherapy agent used, increasing therapeutic effectiveness [7].

To date, many plant-derived anticancer drugs have been implemented as novel drugs (e.g., camptothecin, paclitaxel, and vinblastine) and successfully used in therapy for breast, ovarian, lung, and other types of cancers. Among them, furanocoumarins also play a role as adjuvants in chemotherapy and active anticancer compounds [9]; e.g., photo-activated psoralens have been found to be proapoptotic agents in Jurkat cells [10]. Another furanocoumarin, imperatorin, isolated from *Angelica dahurica*, suppresses the tumor growth, proliferation, and angiogenesis of human colon cancer (HeLa, Hep3B, and HCT116) cells [11].

The general mechanism through which furanocoumarins eliminate cancer cells is based on cell cycle blockage and the initiation of programmed death, like apoptosis or autophagy [9]. Apoptosis dysregulation tends to be a major contributor to both chemoresistance and pathogenesis in many cancers [12]. In our previous study, xanthotoxin (8-methoxypsoralen, 8-MOP) inhibited the growth of several cancer cell lines, with neuroblastoma (SK-N-AS) and metastatic colon cancer (SW620) cells being the most sensitive to this compound. It was found that non-photoactivated xanthotoxin reduced the phosphorylation of AKT^308^ (protein kinase B; phosphoroAkt-thr308), decreased the expression of Bcl-2 (B-cell lymphoma 2 protein), increased the Bax (bcl-like-protein 4; Bcl2-L-4) level, and activated caspases -8, -9, and -3 in both cell lines [13].

The inhibiting of the pro-apoptotic proteins of the Bcl-2 community and blocking of the cytochrome C released from the mitochondria may increase chemoresistance in tumor cells [14,15]. Pro-apoptotic molecules, which act contrarily to this mechanism, may act as chemosensitizers/adjuvants in cancer therapy. Therefore, investigations of the Blc-2/Bax proteins’ expression and a search for intrinsic (mitochondrial) apoptotic pathway mediators may be of great importance. In the present study, we evaluated whether non-UVA-activated 5-methoxypsoralen (bergapten, Figure 1) was active against tumor cell lines such as those of human colorectal adenocarcinoma (HT-29 and SW620), osteosarcoma (Saos-2 and HOS), and multiple myeloma (RPMI8226 and U266).

This is the first report assessing the effect of non-UV-activated bergapten on the proliferation and migration of selected cancer cell lines, providing new data on the potential role of 5-MOP in inhibiting the growth of treatment-resistant cancer cell lines.

## 2. Results

### 2.1. CPC Isolation and LC–MS Identification of 5-MOP

As a result of several CPC semi-preparative steps, pure bergapten (c.a. 99.0%) was isolated from CPE (crude petroleum ether extract) derived from *P. tauricum* fruits, and its identity and purity were confirmed via HPLC–DAD and LC–MS analyses (ESI-MS [H+] (data presented in the Appendix A). The fragmentation *m*/*z* was C_12_H_8_O_4_, MW 216.0901 ESI-MS: *m*/*z* 217.0974 (100, [M + H]+), 174.0436 (4.64), 146.0440 (2.52) 118.0484 (10.4), with the use of the standard compound and the literature’s data [16], as described previously [17]. The isolated bergapten was used in the biological experiments described in this paper.

### 2.2. Cytostatic and Cytotoxic Effect of 5-MOP

The inhibitory activity of 5-MOP against proliferation and cell viability in various cancer cell lines was evaluated using the MTT and NR assays, respectively. The IC50 values (Figure 2A–C and Appendix A) indicated that 5-MOP had diverse anti-proliferative potential with the tested tumor cells at 96 h. The Saos-2 cells showed the highest sensitivity to the compound (IC50 = 40.05 μM). The HT-29 (IC50 = 332.4 μM), SW680 (IC50 = 354.5 μM), and HOS (IC50 = 257.5 μM) cells were characterized by moderate sensitivity, while RPMI8226 (IC50 = 1272 μM) and U266 (IC50 = 1190 μM) were particularly drug-resistant. Importantly, in the case of the HOS line, bergapten at concentrations up to 100 µM stimulated the proliferation of these cells. The observed effect may be related to, among other factors, increased mitochondrial activity in response to the stressor (in this case, 5-MOP). In the defense reaction, cells can induce many mechanisms, activate protective pathways, including, e.g., kinases’ activation (ERK 1/2), or be associated with an increase in the number of mitochondria [18,19]. However, explaining this phenomenon requires more detailed research in the future.

5-MOP not only exerted differential effects on the growth of tumor cell lines but also suppressed their survival to varying degrees.

As demonstrated in Figure 3A–C, the 48-h incubation with bergapten in a 1% medium induced a dose-dependent decrease in cell viability. As indicated in the NR results, the drug-treated Saos-2 culture showed the highest percentage of damaged cells that lost their ability to take up neutral red as a result of cell membrane rupture. In this line, even the lowest dose (6.5 μM) already caused a statistically significant cytotoxic effect. In contrast to the Saos-2 cell line, the invasive (osteosarcoma) HOS cells were less susceptible to the above effect. A statistically significant reduction in the number of viable cells in their culture was achieved at the dose of 50 μM. Bergapten was also found to exhibit relatively low toxicity against the HT-29 and SW680 cell cultures, in which the considerable inhibitory action of 5-MOP was achieved at doses of 100–400 μM. Moreover, in the range of the concentrations tested, this compound appeared to be a slightly stronger cell death inducer in the multiple myeloma cell lines than in the colon cancer cell lines. The threshold concentration required to exert a relevant effect in the RPMI8226 and U266 cells was 50 μM.

Since 5-MOP, in some cancer cell cultures, exhibited more pronounced anti-tumor potential at higher concentrations, we additionally examined the survival rate of non-cancerous human skin fibroblasts (HSF) and hFOB osteoblast cells after 48 h of treatment. As assayed via NR, the low and medium concentrations of bergapten were non-toxic to the normal cells (Figure 3D). Its toxicity was induced only via doses of 200 and 400 μM. When administered at the maximum concentration, it reduced the number of viable HSF and hFOB cells by 26.4% and 35.1%, respectively. All these results allowed the conclusion that, depending on the growth type, origin, and invasive phenotype, neoplastic cells may show diverse responses to 5-MOP treatment. Compared to most of the cancer lines tested, the normal cells exhibited weaker reactivity to the same concentrations used, thus providing evidence of the selective anti-tumor potential of this furanocoumarin.

### 2.3. 5-MOP Decreases the Motility of Cultured Cancer Cells

To check whether 5-MOP affected the cell migration of adherent cancer cell lines (Saos-2, HOS, HT-29, and SW-620) the cells were treated for 24 h with the selected concentrations, and then a wound closure assay was carried out. As shown in Figure 4A,B, bergapten (50–200 µM) reduced the ability of the cancer cells to migrate to the scratch area to varying degrees. As regards the most sensitive Saos-2 cells, the inhibitory effect of 5-MOP was even seen at the concentration of 50 µM and became stronger with a further increase in its concentration. In the case of the HT-29 cells, statistically significant changes in their mobility were noted only at the concentration of 100 µM. In the cases of the other tumor cell lines, statistically insignificant decreases or unchanged migratory activity were observed. 

### 2.4. 5-MOP Induces Alterations in Cell Cycle Distribution

Since the Saos-2 cells were the most sensitive to 5-MOP (with the lowest IC50 values at 96 h) compared to the other cell lines applied, further mechanistic studies were performed using these cells as an in vitro model. To determine whether the anti-proliferative potential of 5-MOP was related to cell cycle perturbation, the content of propidium-iodide-labeled DNA in the treated and untreated (control) Saos-2 cells was measured via flow cytometry. As shown in Figure 5A,B, the 48-h incubation with bergapten resulted in a dose-dependent reduction in the G1 phase cells. At the concentration of 200 μM, the G1 fraction decreased by about 39% compared to the control. The treatment also resulted in a weak decrease in the number of S phase cells (by 3.6% at 200 μM) and a clear increase in the G2 phase cells. However, the greatest number of Saos-2 cells arrested in the G2 phase was observed after incubation with 50 μM. In addition to the changes in the G1, S, and G2 phases, the cultures exposed to high doses of bergapten (100 and 200 µM) showed a rapid increase in the fraction of sub-G1 cells with reduced DNA content. The massive accumulation of cells in the sub-G1 phase might, therefore, indicate the effective induction of apoptotic cell death. The results suggest that 5-MOP-mediated cell death could be attributed to the induction of G2/M phase arrest and apoptosis.

### 2.5. 5-MOP Induces Apoptotic Cell Death in Saos-2 Cells

To further clarify the ability of bergapten to trigger cell destruction via apoptosis and/or necrosis mechanisms, the Saos-2 cells were treated with the compound, and then a quantitative evaluation of these phenomena was performed using flow cytometry. The 48-h incubation with 5-MOP elevated the apoptosis rate in the osteosarcoma culture in a dose-dependent manner (Figure 6A,B). The percentage of total apoptotic populations (early + late) increased from 32.30% (at 50 µM) to 44.50% (at 100 µM) compared with the untreated group (3.56%). It was also found that none of the concentrations used caused a strong increase in the number of necrotic (propidium-iodide-labeled) cells. In response to the highest dose of 5-MOP, only 4% of cells underwent necrosis. These results indicate that the promotion of programmed cell death was mainly responsible for the decreased cell viability upon the administration of the 5-MOP treatment.

The pro-apoptotic effect of bergapten was additionally confirmed via the measurement of the active form of executioner caspase-3 and initiator caspase-9 in the Saos-2 cell line. Caspases are cysteine-aspartic proteases whose activation is part of a complex process that consistently leads to the degradation of cellular components in cells that die via apoptosis. Caspase-9 is a representative of initiator caspases, while caspase-3 is one of the key executive enzymes [20]. As shown in Figure 7A,B, treatment with 5-MOP induced a concentration-dependent rise in the number of osteosarcoma cells with active caspase-9. Compared to the control culture (4.50%), the percentage of caspase-3 (cleaved) positive cells at the 25 µM, 50 µM, and 100 µM 5-MOP concentrations was 15.84%, 21.30%, and 28.48% respectively. Along with the induction of caspase-9, there was also a significant increase in caspase-3 activity in the treated cells (Figure 7C,D). The level of its active, cleaved form increased 11-fold compared to the control when the highest (i.e., 100 µM) concentration of 5-MOP was applied. The flow cytometry results presented here clearly support the idea that bergapten is a potent inducer of caspase-dependent apoptosis.

### 2.6. 5-MOP Affects the Mitochondrial Membrane Potential and Modulates Bax and Bcl-2 Protein Expression

The decline in mitochondrial transmembrane potential is an irreversible event preceding the induction of the caspase cascade and activation of nuclei (enzymes capable of breaking phosphodiester bonds between nucleic acid nucleotides, resulting in the disintegration of the cell nucleus). Therefore, the determination of mitochondrial membrane potential (MMP and ΔΨ_m_) allows the identification of early-apoptotic cells in cell cultures upon drug administration. As shown in Figure 8A,B, bergapten led to significant mitochondrial depolarization at both concentrations (50 and 100 µM) tested compared to the control. In the presence of 100 µM, it was detected that almost 52% of the treated cells underwent a loss of mitochondrial membrane potential.

Because 5-MOP was able to activate the mitochondrial caspase-dependent apoptotic pathway (by interfering with the physiological function of mitochondria and inducing caspases), we investigated its impact on the expression of Bcl-2-family proteins (regulators of the process by which mitochondria participate in apoptosis, known as the intrinsic apoptosis pathway). The analysis of the Western blot results (the uncropped Western blot images are presented in Appendix A) revealed pronounced changes in the levels of both Bcl-2 and Bax proteins in the 5-MOP-treated cells. As shown in Figure 8C–E, the amount of Bax protein gradually increased as the concentration of 5-MOP increased, while the level of Bcl-2 protein declined, reaching the greatest decrease at the concentration of 100 µM of 5-MOP.

### 2.7. Reducing the Level of AKT Phosphorylation Enhances the Pro-Apoptotic Effect of 5-MOP

To test whether the PI3K/AKT pathway (known to have a pro-survival and mitogenic function) could be inhibited via 5-MOP, the level of total AKT and AKT phosphorylation (phosphoAKT-thre308, AKT^308^, and *p*-AKT, a master signal transducer of the signaling cascade) was analyzed via quantitative ELISA. The 5-MOP-induced suppression of AKT phosphorylation for each treatment was determined by calculating the ratio of AKT phosphorylated at Thre380 to the total AKT (pAKT/tAKT). As shown in Figure 9A, the 24-h incubation with 5-MOP decreased the amount of *p*-AKT^308^ in the treated cells in a concentration-dependent manner. At a dose of 100 µM, the activation of this kinase was reduced by 54.07% compared to the control cells. To further elucidate the influence of 5-MOP on the PI3K/AKT pathway, apoptosis was estimated. The Saos-2 cells were cultured for 48 h with 10 µM LY294002 (a selective inhibitor of PI3K) in the presence or absence of 5-MOP (50 µM). The results obtained from the flow cytometry analysis revealed that apoptosis was substantially increased in the cells co-incubated with 5-MOP and the PI3K inhibitor compared to the cells treated with 5-MOP alone or with the PI3K inhibitor alone (Figure 9B,C). These data may, therefore, suggest that 5-MOP can exert anti-osteosarcoma activity by hindering of PI3K-AKT cascade in cancer cells.

## 3. Discussion

Nowadays, attention is increasingly being paid to natural substances in cancer therapy. Due to their easy availability, efficacy, and safety of use, they can be a valuable support for chemotherapeutics. Naturally occurring products with proven anti-cancer effects include, e.g., resveratrol, found in red wine, curcumin, derived from *Curcuma longa* (Zingiberaceae), botulin, isolated from white birch species, and statins produced via the fungus *Aspergillus terreus* [21,22,23]. Compared with conventional cytostatic agents (e.g., methotrexate, cisplatin, or 5-FU), known to induce serious side effects, most phytochemicals exhibit relatively high tumor selectivity (a property that keeps healthy cells alive), and what is more, they can sensitize cancer cells to chemo- and/or radiotherapy [24,25,26]. One of the groups of natural biomolecules that has recently attracted the attention of researchers due to its anticancer potential is furanocoumarins, which include 5-methoxypsoralen (bergapten; 5-MOP). This compound, found in many plants from the Apiaceae, Rutaceae, and Moraceae families [27], is a component of citrus essential oils, including bergamot oil (*Citrus bergamia*), which is present in grapefruit juice (*Citrus paradis*) and has been isolated from various parts (mainly the fruits and roots) of the *Heracleum, Angelica*, and *Peucedanum* species (family Apiaceae). Bergapten performs various pharmacological activities, including chemopreventive, antiproliferative, anti-inflammatory (reducing cyclooxygenases—COX—tumor necrosis factor-α, and interleukin-6 and NO release), antioxidant, antibacterial, hepatoprotective, and antiosteoporotic effects [27,28,29,30]. As a photochemotherapeutic agent, it can be applied to patients with hyperproliferative skin disorders, such as psoriasis or vitiligo [31,32]. Unfortunately, prolonged overexposure to UVA-excited psoralens (PUVA therapy—psoralens and UVA) may augment the risk of basal cell, squamous cell, and melanoma skin cancers [33,34]. Although many aspects of PUVA therapy have been widely investigated [35,36], the effects and mode of 5-MOP action, regardless of UV exposure, still need to be fully explored in in vitro cultured cancer cells of different origins and genotypes. Therefore, in the present work, we evaluated the anti-tumor potential of non-UV-activated 5-MOP in various human cancer cell lines, including HT-29 and SW620 (colorectal adenocarcinoma), U266 and RPMI8226 (multiple myeloma), and Saos-2 and HOS (osteosarcoma). It is worth mentioning that all these lines, except for one (HT29), had not previously been tested in the assessment of bergapten toxicity. Our in vitro study showed diverse concentration-dependent cytostatic and cytotoxic effects of 5-MOP on the applied cell lines. The IC50 values calculated on the basis of the MTT (3-(4,5-Dimethylthiazol-2-yl)-2,5-Diphenyltetrazolium Bromide) test results revealed that the Saos-2 cell line was the most sensitive to bergapten (40.05 µM), while the HOS line was almost six times less responsive. The colon cancer cells were characterized by moderate sensitivity (332 µM for HT-29 and 345.5 µM for SW625), and the multiple myeloma cells RPMI (1272 µM) and U266 (1190 µM) exhibited the weakest reactivity. Since bergapten was found to have no unique preference for a specific type of tumor tissue, we assumed that the genetic profile of the lines tested may have determined its differential potency. Previous in vitro studies have also confirmed the varied antitumor action of non-UVA-activated 5-MOP against other types of tumor cells. Fujioka et al. [37] showed the antiproliferative activity of bergapten on MK-1 (gastric cancer), HeLa (cervical cancer), and B16F10 (murine melanoma) cells, with IC50 values of 193.0 μM, 43.5 μM, and >462.0 μM, respectively. As in the present study, Girennavar et al. [38] demonstrated the low toxicity of bergapten to the HT29 line in both 24- and 48-h cultures. However, its slightly stronger effect on these cells was reported by Um et al. [39]. In another published work, bergapten (with or without UV irradiation), even at low concentrations (<25 μM), appeared to be a potent inhibitor of the proliferation of cultured breast cancer cells, including MCF-7, SKBR-3, and ZR-75 [40], and U87MG glioblastoma cells [41]. Bergapten was also evidenced to suppress the viability of A549 and NCI-H460 non-small-cell lung cancer cells [42]. In contrast to the above-mentioned cancerous cells, in the hepatocellular carcinoma J5 cell line, the lethal effect of bergapten was revealed in the range of very high (millimolar) concentrations of 5–50 mM [24].

Since the cellular toxicity of anticancer substances for healthy tissues is a major factor limiting their therapeutic use, we simultaneously determined (via an NR assay) the impact of 5-MOP on the survival of normal human HSF and hFOB cells. Only high doses of 5-MOP (200 µM and 400 µM) resulted in statistically significant cell death through plasma membrane damage. These findings are consistent with the results of other in vitro experiments, in which malignant cells were more vulnerable than their normal counterparts to psoralens (5-MOP or 8-MOP) [13,38,43,44]. In addition, no animal studies or clinical trials conducted to date have documented acute side reactions after the oral or topical administration of 5-MOP [27,28,29,45]. Based on our results and the literature’s data, we supposed that the intracellular target sites and/or signaling pathways in pathological cells are more susceptible to the effects of 5-MOP than in normal cells. Given the low potential toxicity and tumor selectivity, this non-UV-irradiated methoxyfuranocoumarin may serve as a good chemotherapeutic agent and/or an adjuvant in cancer treatment.

Because the Saos-2 cells were the most severely affected by bergapten, this cell line was used in further studies to establish the cellular and molecular mechanisms of its cytotoxic action. As demonstrated in our screening studies, the reduced metabolic activity and increased membrane permeability of the Saos-2 cells exposed to 5-MOP may be a sign of mitochondrial dysfunction and/or impaired ability to divide and survive. It has recently been reported that 5-MOP induces death in cancer cells and evokes growth inhibition by interfering with cell cycle progression and the expression of proteins related to the cell cycle and apoptosis control [24,41,46]. Consistently, our results demonstrated that the 5-MOP treatment induced G2/M cell cycle arrest (especially at 50 µM) and elevated the number of cells with apoptosis-specific hypoploid DNA content. The cell cycle inhibition via 5-MOP in the G2/M phase with subsequent apoptosis induction was also shown by Lee et al. [24] and Guo et al. [41] in hepatoma (J5) and glioma cells (U87MG), respectively. In turn, in a study conducted by Lin et al. [46] on colon cancer cells (DLD-1 and LoVo), bergapten treatment triggered p53-mediated biochemical events associated with cycle blockage in the G0/G1 phase and cellular apoptosis. The same observations were made by Panno et al. for breast cancer cell lines (MCF-7 and SKBR-3) with wild-type p53 [35]. In the case of the osteosarcoma Saos-2 cells, the mitotic delay and viability suppression could not be driven by p53, as these cells lost their normal function due to gene mutation. Therefore, we conclude that molecular targets other than p53 may be implicated in the sensitivity of the Saos-2 cells to bergapten.

Via FACS (fluorescence-activated cell sorting) dot plot analysis using AnnexinV/FITC (Annexin V and fluorescein) and propidium iodide double-staining, we definitively confirmed that apoptosis was the predominant form of cell death in the Saos-2 cultures after the 5-MOP treatment. The rate of dead cells assessed via flow cytometry was consistent with that detected using the NR assay. We found that bergapten substantially increased the amount of apoptotic cells (in the early and late apoptosis stages) in a dose-dependent manner but had no significant impact on necrosis. These results supported the previous findings on the apoptogenic activity of this furanocoumarin reported by other authors [41,42,47,48]. Some in vitro studies showed that 5-MOP induced an autophagic or necrotic response in addition to apoptosis [24,49]. In breast cancer models (MCF-7 and ZR-75), bergapten was shown to exert an anti-survival effect through the process of autophagy, as evidenced via the PTEN-dependent up-regulation (PTEN—dual-specificity protein phosphatase, dephosphorylating tyrosine-, serine-, and threonine-phosphorylated proteins) of its key protein markers, i.e., Beclin1 (Bcl-2 interacting protein), PI3KIII (phosphatidylinositol 3-kinase), UVRAG (UV radiation resistance-associated gene protein), and AMBRA (actin binding Rho-activated protein). More interestingly, this event preceded the onset of apoptosis in these cells [49]. Whether these autophagic changes are specific only to certain types of tumors or can occur in osteosarcomas due to the 5-MOP stimulus still remains to be clarified.

Drug-induced apoptosis most often occurs via the initiation of intrinsic and/or extrinsic signaling [20]. The extrinsic pathway is triggered via the interaction of membrane death receptors (Fas, TNFR, and TRAIL) with their specific ligands (TNF, FasL, and TRAIL DR) and is related to the active function of initiator caspases (-8 and -10), which regulates executioner caspase-3 processing. Conversely, the intrinsic pathway, also termed the mitochondrial cascade, is triggered through the release of cytochrome c and Apaf-1 proapoptotic factors from the mitochondrial membrane to the cytosol. Unleashed cytochrome c acts as a caspase activator; it indirectly stimulates the activation of initiator caspase-9, followed by caspase-3. Caspase-3, as the most critical effector in death machinery, is responsible for the morphological and biochemical changes characteristic of apoptosis [20,50]. Moreover, intrinsic signaling is relatively dependent on mitochondrial membrane-coupled Bcl family members, including Bax and Bcl-2, which are pro- or anti-apoptotic molecules, respectively [51]. Consistent with some previous work [42,46], our mechanistic experiments conducted on the Saos-2 cells indicated that the mitochondrial pathway played a central role in 5-MOP-mediated apoptosis. This was supported by the following observations: alterations in MMP (mitochondrial membrane potential), Bax, and Bcl-2 protein expression and caspase -9 and -3 activity. The proapoptotic Bax (whose level was markedly elevated after the 5-MOP treatment) could trigger the destabilization of the mitochondrial outer membrane via the formation of megapores. This successively led to the loss of mitochondrial potential (as evidenced via the reduced fluorescence intensity of the cells stained with the ΔΨ_m_-sensitive dye DIOC_6_) and the cascade activation of both caspases (-9 and -3), i.e., events that ultimately caused the Saos-2 cells’ suicidal death. While most results point to the mitochondrial pathway as that underlying the anticancer effects of this phytochemical, only a few findings suggest the contribution of the receptor pathway as well. The ability of bergapten to generate apoptotic responses through the up-regulation of not only caspase-9 but also caspase-8 has, indeed, been revealed in MCF-7 (a human breast cancer cell line with estrogen, progesterone, and glucocorticoid receptors) and SKBR-3 cells (a human breast cancer cell line) [35], thus highlighting the possible importance of these two death tracks in its action.

Various cell signaling pathways are non-functional or overactive in cancer cells; hence, these cells divide uncontrollably and are resistant to death signals. Particularly, the phosphatidylinositol 3-kinase (PI3K)/AKT axis is believed to be one of the most severely dysregulated in many types of human malignancies, including colorectal cancer cells and osteosarcoma [52]. Through a complex network of interactions with various downstream target molecules, this pathway is responsible for almost all aspects of tumor development: abnormal proliferation, cell cycle progression, invasion, metastasis, or angiogenesis [53]. AKT (serine/threonine kinase or protein kinase B, PKB) is recognized to be a major oncogenic regulator of this pathway, and its consistent overexpression/activation observed in tumor cells is related to blocking of drug-induced apoptosis via multiple mechanisms, such as amplifying the effect of anti-apoptotic modulators (Bcl-2 and survivin) and counteracting the functions of pro-apoptotic (Bax, Bad, and p53) proteins [54,55]. Furthermore, numerous data have demonstrated that PI3K/AKT/mTOR (mTOR: mammalian target of rapamycin kinase) inhibitors may have strong suppressive activity against osteosarcoma progression [56]. For this reason, the suppression of AKT signaling is regarded as a potential therapeutic strategy for the elimination of malignant cells [54].

Our data demonstrated that 5-MOP inhibited the phosphorylation of AKT^308^ in the osteosarcoma cell line model within 24 h. In addition, the co-incubation of Saos-2 cells with bergapten and LY294002 enhanced the susceptibility of these cells to apoptosis execution. LY294002 (2-(4-morpholinyl)-8-phenyl-4H-1-benzopyran-4-one) is a known PI3-kinase inhibitor that induces apoptosis in various cancer cell models by down-regulating AKT/PKB activation [57,58]. According to the literature’s data, combined treatment (a PI3K or AKT inhibitor with the tested compound) is an alternative (inexpensive) approach to possibly explaining whether the cytotoxicity of the drug is related to the inactivation of the PI3K/AKT pathway [59,60]. When we used the tested compounds separately (the PI3K inhibitor or 5-MOP), in both cases, there was an increase in apoptosis in the Saos-2 culture as compared to the control. However, after the simultaneous application of LY294002 and 5-MOP, we could observe a two-fold increase in the apoptosis rate compared to each of the tested groups. This, therefore, suggests that the AKT signaling molecule could, in part, mediate the proapoptotic function of 5-MOP in the Saos-2 cells. The capability of this compound to interrupt signal transduction along the PI3K/AKT cascade was also substantiated via several other groups [35,40,45,46]. In breast cancer, hepatocarcinoma, and gliomas, the anti-survival effects of 5-MOP were attributed to PI3K downregulation, as well as its downstream AKT and mTOR effectors [35,41,46]. Based on these facts, it seems that various transmitters of the PI3K/AKT pathway may serve as critical modulators of the anti-cancer potential of non-photoactivated bergapten. The intensification of the apoptosis process via the combined administration of LY294002 and 5-MOP may only suggest, though not fully confirm, the direct involvement of the PI3/AKT pathway in the pro-apoptotic effect of 5-MOP; therefore, more detailed functional studies are necessary to clarify this issue.

Cell invasion is one of the main features of metastatic cancer. The ability of cells to metastasize is determined by their increased migratory activity. Through the inhibition of this potential, the aggressive phenotype of malignant cancer cells can be inhibited, thus limiting their progressive growth. In the current study, we also determined the antimigratory properties of 5-MOP [61,62]. We found that this compound exhibited cell-type-dependent cell migration inhibitory activity. A marked and statistically significant change in cell motility was noted in the Saos-2 and HT-29 lines. In the SW680 line, these changes were barely visible, while in the other cells tested, the migration was at the level of the control. In agreement with our observation, Zhang et al. [63] revealed that bergapten-containing products derived from *Ficus carica* were able to impair the migration ability of the breast cancer cell line MDA-MB-231 by modulating the expression of crucial factors in the invasion process, such as MMP2 (metalloproteinase 2), TIMP-1, and TIMP-2 (tissue inhibitors of metalloproteinases 1 and 2). The experiments may, therefore, suggest an anti-metastatic nature of this molecule. Given the limited number of studies on this topic, additional research is needed to clarify the mechanisms underlying this action.

In summary, the present findings confirmed the attractive anticancer profile of non-UV-activated bergapten, the methoxyfuranocoumarin derived from a plant source. The multidirectional action of this compound (anti-growth, anti-survival, anti-migrative, and proapoptotic), especially against the Saos-2 line, is an important premise to undertake in vivo studies and, in the future, to consider in clinical trials to verify its antineoplastic potential, as well as its possible side effects. If promising results are obtained, bergapten can be considered a promising substance of natural origin for chemoprevention and/or the therapy of cancers sensitive to it.

## 4. Materials and Methods

### 4.1. Solvents and Chemicals

Acetonitrile and methanol (HPLC-gradient-grade) were purchased from J.T. Baker (Deventer, The Netherlands); *n*-heptane, ethyl acetate, and petroleum ether (PE) were analytical-grade (POCH (Gliwice, Poland). Ultrapure water (18.2 MΩ), as obtained from a Simplicity (Millipore, Molsheim, France) purification system, was used. For the LC-MS experiments, the water and acetonitrile were of LC–MS-grade (J.T. Baker, Atlanta, GA, USA). Bergapten (purity ≥ 98%; assayed by HPLC, Sigma, St. Louis, MO, USA) was used as a reference substance.

### 4.2. Plant Material

Mature fruits of *Peucedanum tauricum* (Apiaceae, MB-PT/01/2014) were collected from the Botanical Garden of Maria Curie-Skłodowska University (51°16′ N, 22°30′ E, 200 m AMSL, Lublin, Poland) and were identified by plant taxonomist Krystyna Dąbrowska, MSc. Before use, the fruits were dried in a dark place at room temperature (humidity < 30%, temp. c.a. 25 °C).

### 4.3. Isolation, Purification, and Identification of 5-MOP

#### 4.3.1. Extraction of the Plant Material

The dried plant material was pulverized, macerated with petroleum ether (24 h), and then extracted exhaustively with the same solvent (25 g/250 mL). The extract was concentrated, and after evaporation of the solvent, crude petroleum ether extract (CPE) was obtained and subjected to the next steps of analysis and separation.

#### 4.3.2. CPC Isolation of 5-MOP

The isolation of 5-MOP from CPE was performed with the use of optimized centrifugal partition chromatography (CPC), as described previously [17]. Briefly, the CPC instrument employed in the present study was a model Armen SCPC-250-L (Armen Instrument, Saint Ave, France) equipped with an SCPC-250 Teflon column with a total capacity of 250 mL that was integrated with a gradient flow pump and a UV lamp (Flash 06S DAD 600) operating at various wavelengths. The CPC apparatus was equipped with a manual injector with a 10 mL sample loop. The system was controlled via the software Armen Glider CPC, V5.0a.05.

In each separation run, the column was first filled entirely with the stationary phase at the flow rate of 20 mL/min (500 rpm and 20 min). Then, the upper organic phase was pumped into the column. The sample 150 mg of the CPE (semi-preparative scale), dissolved in the mixed two phases (4 mL of each and, finally, 8 mL), was injected through the injection valve after the equilibration of the column. Effluent from the column was continuously monitored via UV detection (at 254 nm and 320 nm, suitable for coumarin detection), and the peak fractions (32 mL of each) were collected using an automatic fraction collector. Separation was performed in the ascending mode with the use of the optimized biphasic solvent system (n-heptane-ethyl acetate-methanol-water; 5:2:5:2, *v*/*v*/*v*/*v*). The flow rate was 3 mL/min, and the rotation speed was 1600 rpm.

#### 4.3.3. LC–MS Identification of 5-MOP

The fractions of CPE purified via CPC were analyzed using HPLC/DAD (the selectivity of the compounds was confirmed via DAD spectra in the range of 190–400 nm). The HPLC equipment used was an Agilent 1100 system (a G1311A QuatPump, a G1315BDAD, an ALS Therm G1330B thermostat, a Rheodyne injection valve with a 20 μL loop, an ALS G1329A autosampler, and a G1322A membrane degasser), equipped with a Zorbax Eclipse XDB C18 column (250 mm × 4.6 mm ID, 5 μm, Agilent Technologies, Santa Clara, CA, USA) controlled using an Agilent HPLC OpenLab CDS ChemStation. The flow rate was 1 mL/min, the temperature was 25 °C, and the gradient mobile phase methanol (A) in water (B) was as follows: 0–5 min, 60% A in B; 5–20 min. from 60 to 80% A in B; 20–30 min. 80 to 85% A in B; and post time 10 min at 60% A in B. The isolated compound and corresponding external standard were carefully weighed, and a stock solution of 1 mg of each was dissolved in 10 mL of methanol and then analyzed. All the calculations concerning the quantitative analysis were performed with a calibration curve for the targeted compound (5-MOP) via the measurement of peak areas at λ = 320 nm. The analysis was monitored in each case via DAD at 320 nm for coumarin compounds and at 254 nm and 360 nm for impurities.

The identity of isolated bergapten was confirmed in MS experiments performed with the use of a 6210 TOF LC/MS mass spectrometer (Agilent Technologies, Santa Clara, CA, USA) with MassHunter Software v. 10.0 (Agilent Technologies, Santa Clara, CA, USA). The XBridge^TM^ Shield RP18 (100 × 2.1 mm ID, 3.5 µm) column (Waters; Milford, MA, USA) and the mobile phase (flow 0.2 mL/min; pH 4.5; HCOONH_4_)—A: 1% MeCN; B: 95% MeCN; 1–15 min, 30% B; 15–25 min, 45% B; and 25–35 min, 85% B—were used, and the post time took 10 min. The analysis was performed in the mass range of 100–1000 *m*/*z*. A fragmentor of 215 V (positive ion mode), a capillary voltage of 4000 V, a skimmer of 65 V, nebulizer pressure of 35 psi, agas temperature of 350 °C, and a nitrogen flow of 10 L/min were used. In each case, 2 µL of the sample was injected.

### 4.4. Biological Studies

#### 4.4.1. Cell Cultures

The cancer cell lines analyzed in this study were selected to be of different origin, and all of them represented chemotherapy-resistant cancers, such as osteosarcoma, colorectal adenocarcinoma, and multiple myeloma (the data presenting characteristics of the investigated cell lines are in the Appendix A, [64,65,66,67,68]).

The human tumor cell lines for colorectal adenocarcinoma derived from the metastatic side’s lymph node, Dukes’s type C (SW620), osteosarcoma (Saos-2 and HOS), multiple myeloma (U266 and RPMI 8226), and human normal osteoblasts (hFOB 1.19) were purchased from ATCC, Manassas, VA. Human colorectal adenocarcinoma (HT-29) was obtained from the Institute of Immunology and Experimental Therapy (Polish Academy of Sciences, Wroclaw, Poland). The normal human skin fibroblasts (HSF) were a laboratory strain established using an outgrowth technique from the skin explants of healthy subjects (with written informed consent). The U266 and RPMI8226 cells were maintained in an RPMI1640 medium, the Saos-2 and HT-29 cells in McCoy’s 5A Modified Medium, SW620 in an L-15 medium, HOS and HSF in a MEM medium, and hFOB 1.19 in a 1:1 mixture of DMEM without phenol red and Ham’s F12 medium. All media were purchased from Sigma Aldrich (St. Louis, MO, USA) and supplemented with 10% fetal bovine serum (FBS; PAN-Biotech GmbH, Aidenbach, Germany). All media (except the medium for hFOB) also contained 100 U/mL of penicillin and 100 μg/mL of streptomycin (Sigma Aldrich). The hFOB1.19 cells were cultivated at a permissive temperature of 34 °C, and all the other cultures at 37 °C, in a humidified atmosphere of 95% air and 5% CO_2_. Stock solutions of isolated 5-MOP were prepared in DMSO (Sigma-Aldrich) and stored at 4 °C until use. The DMSO concentration did not exceed 0.1% (*v*/*v*). The compound was diluted in an appropriate culture medium (with 1% or 10% FBS) immediately before use.

#### 4.4.2. Proliferation Assay

The antiproliferative potential of bergapten against cancer cells was evaluated after 96-h exposure using the colorimetric MTT assay, and the IC_50_ values were calculated as described previously [69]. Briefly, the cells were seeded into 96-well plates (Nunc, Rochester, NY, USA) at densities of 3 × 10^3^ (HT-29), 5 × 10^3^ (SW620), 4 × 10^3^ (Saos-2 and HOS), and 2 × 10^4^ (U266 and RPMI 8226) cells/well in 100 μL of a complete growth medium. After overnight attachment, the culture media were removed, and the cells were treated with an increasing concentration of 5-MOP. The untreated cells (0 μM) were cultivated with an appropriate cell culture medium with 10% of FBS. Next, the cells were incubated with an MTT solution (5 mg/mL) (Sigma-Aldrich) for 3 h, and blue formazan crystals were dissolved in a sodium dodecyl sulfate buffer (10% SDS in 0.01 N HCl) overnight. The absorbance was measured at a wavelength of 570 nm using the E-max Microplate Reader (Molecular Devices Corporation, Menlo Park, CA, USA). The absorbance of the control wells was taken as 100%, and the results were expressed as a percentage of the control.

#### 4.4.3. Cytotoxicity Assay

The toxicity of bergapten against normal and tumor cells was determined using the neutral red (NR) uptake assay. This test is based on the ability of living cells to incorporate the supravital, cationic dye into lysosomes via an active transport. In contrast to vital cells with intact cell membranes, damaged or dead cells do not accumulate NR. Briefly, the optimized amounts of cells were seeded in 96-well plates. After overnight incubation at 37 °C or 34 °C, the cultures were subjected to 48 h of exposure to serial dilutions of 5-MOP, prepared in a medium with 1% FBS (growth restriction). The cells were then incubated for 3 h with a medium containing 0.4% NR (Sigma-Aldrich). The plates were subsequently fixed with 1% CaCl_2_ in 4% paraformaldehyde (Sigma-Aldrich), the incorporated dye was extracted from each well with 50% ethanol, and the absorbance of the samples was determined at a wavelength of 540 nm using an E-max Microplate Reader (Molecular Devices Corporation, Menlo Park, CA, USA).

#### 4.4.4. Western Blot Method

Immunoblotting analysis was carried out as previously described [70]. Briefly, the sub-confluent cell cultures were incubated in a medium with 1% FBS that contained bergapten (25, 50, or 100 µM). The untreated control cells (0 μM) were cultured without the tested compound. After 48-h exposure, the cells were harvested, lysed on ice in a RIPA buffer supplemented with a protease inhibitor cocktail (Sigma-Aldrich), and centrifuged (14,000 rpm for 10 min at 4 °C). The total protein concentration was estimated via incubation with a bicinchoninic acid protein assay reagent (Pierce^®^ BCA Protein Assay Kit; Thermo Fisher Scientific, Rockford, IL, USA). Samples with equal protein concentrations were electrophoresed and electrotransferred to polyvinylidene difluoride (PVDF) membranes. After blocking with 5% non-fat milk in Tris-buffered saline and 0.05% Thermo Scientific Tween 20 detergent, the membranes were incubated overnight at 4 °C with anti-Bcl-2 and anti-Bax primary antibodies (1:1000, Santa Cruz Biotechnology, Santa Cruz, CA, USA) and then with horseradish-peroxidase-(HRP)-conjugated secondary antibodies (1:2000, 1 h RT, Santa Cruz Biotechnology). Antigen–antibody complexes were detected using chemiluminescent reactions and analyzed by means of a Molecular Imager^®^ ChemiDocTM XRS+ (Bio-Rad Laboratories, Hercules, CA, USA) equipped with Image LabTM Version 3.0 Software for the measurement of protein band intensity. The blots were reprobed with antibodies against *β*-actin (1:500, Sigma-Aldrich) used as a load control. The changes in the protein level were expressed as a % of the control (100%).

#### 4.4.5. Cell Cycle Analysis

Cell suspensions (3 × 10^5^ cells/mL) were plated on 6-well plates (Nunc) and left overnight. Next, the medium was changed to a new one containing 10% FBS, and the cells were treated for 48 h with a serial dilution of 5-MOP (25, 50, and 100 μM). The cell cycle progression in the treated and control cell groups was assessed in a flow cytometer using the PI/RNase Staining Buffer (BD Biosciences, BD Pharmingen™, San Diego, CA, USA), as described previously [71].

#### 4.4.6. Flow Cytometry

The quantitative analysis of apoptotic and necrotic cells was determined using an Annexin V-fluorescein isothiocyanate (FITC)/propidium iodide (PI) apoptosis kit (BD Biosciences, BD Pharmingen., San Jose, CA, USA) and the flow cytometry method, as described previously [13]. Briefly, cell suspensions (3 × 10^5^ cells/mL) were plated on 6-well plates (Nunc) and cultivated for one day. Next, the medium was changed to a new one containing 1% FBS, and the cells were treated for 48 h with a serial dilution of bergapten (25, 50, and 100 μM). In some experiments, the cells were exposed to 5-MOP (50 μM), LY294002 (a selective inhibitor of PI3K, 10 μM, Sigma-Aldrich), or a combination of these two compounds. After harvesting and washing with PBS, the cells were stained with FITC-Annexin V and PI and subsequently analyzed using a flow cytometer (BD FACSCalibur) with the CellQuest Pro Version 6.0 software. The fluorescence-activated cell sorting (FACS) technique was also applied to measure the activity of caspase-3 and caspase-9 in the treated or untreated cells. After 48-incubation with 5-MOP, a phycoerythrin (PE) Active Caspase-3 Apoptosis Kit (BD Biosciences, San Jose, CA, USA) and a fluorescein CaspaTag Caspase 9 In Situ Assay kit (Sigma-Aldrich) were used according to the manufacturers’ instructions.

#### 4.4.7. Measurement of the Mitochondrial Membrane Potential

The quantitative measurement of the mitochondria membrane potential (ΔΨ_m_) was carried out via flow cytometry utilizing a mitochondria-specific and voltage-dependent fluorescent probe and 3,3′-dihexyloxacarbocyanine iodide (DiOC6(3) (Sigma-Aldrich). After 24-h incubation with selected concentrations of 5-MOP (50 and 100 µM), the Saos cells were harvested, washed with PBS, and resuspended in a DiOC6 solution (30 nM in PBS). The cell suspensions were incubated in the dark at 37 °C for 15 min before flow cytometric analysis was conducted at the FL1 canal.

#### 4.4.8. ELISA Assay

The quantification of the intracellular levels of total AKT and phosphorylated AKT^308^ in the exposed cells was performed using the PathScan^®^ ELISA kits (Cell Signaling Technology, Danvers, MA, USA), according to the manufacturer’s protocol, as described previously [72]. After 24-h exposure with selected doses of 5-MOP, untreated (control) and treated cell lysates were incubated in a 96-well plate with an antibody cocktail, and following the washing steps, they were treated with a 3,3′,5,5′-tetramethylbenzidine (TMB) colorimetric substrate. At the end of the assay, the optical density was measured using an E-max Microplate Reader (Molecular Devices Corporation, Menlo Park, CA, USA).

#### 4.4.9. Cell Migration Assessment

The influence of 5-MOP on tumor cell motility was evaluated in a wound healing assay. Cells (5 × 10^5^ cells/mL) were seeded in culture dishes (4 cm in diameter, Nunc) in 2 mL of complete medium. After 24-h, the monolayer of the cells was scratched with a pipette tip (P300) to create one linear wound. The cells were rinsed with PBS and covered with a fresh growth medium supplemented with 5-MOP (50–200 µM) or without furanocoumarin (control cells). The next day, the cultures were stained according to the May–Grünwald–Giemsa method [73] and observed under an Olympus BX51 System Microscope (Olympus Optical Co., Ltd., Tokyo, Japan). The inhibition of cell migration was estimated using the method in which the average width of the wounds in control cultures and in cultures with 5-MOP was measured on micrographs. Micrographs were prepared using the AnalySIS^®^ software v. 510_UMA_cellSens 1.13 (Soft Imaging System GmbH, Münster, Germany).

### 4.5. Statistical Analysis

Statistical analyses were conducted using GraphPAD Prism 5 (GraphPAD Software Inc., San Diego, CA, USA). The data were analyzed via a one-way ANOVA, followed by Dunnett’s or Tukey’s multiple comparison tests. The values were represented as means ± SDs, and *p* values < 0.05 were considered statistically significant.

## Figures and Tables

**Figure 1 ijms-24-15555-f001:**
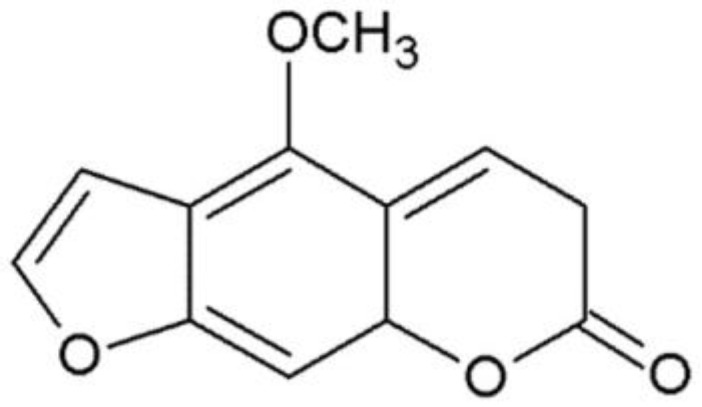
The chemical structure of 5-methoxypsoralen (bergapten; 5-MOP).

**Figure 2 ijms-24-15555-f002:**
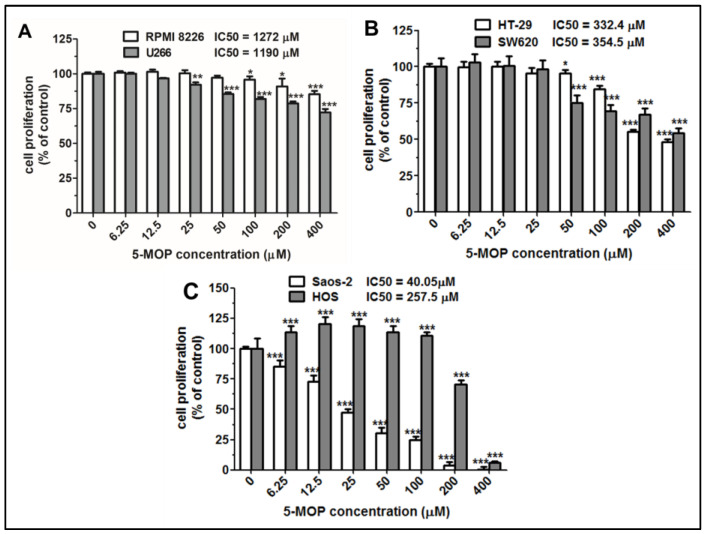
The effect of 5-MOP treatment on the proliferation of different cancer cells. (**A**) Multiple myeloma cells: RPMI 8226 and U266; (**B**) colon cancer cells: HT29 and SW620; (**C**) osteosarcoma cells: Saos-2 and HOS. The cells were cultured alone or in the presence of increasing concentrations of 5-MOP. Cell proliferation was estimated using an MTT assay at 96 h. Each bar represents the mean ± standard deviation (SD) of three independent experiments in comparison to the control (DMSO). One-way ANOVA test; * *p* < 0.05, ** *p* < 0.01, and *** *p* < 0.001.

**Figure 3 ijms-24-15555-f003:**
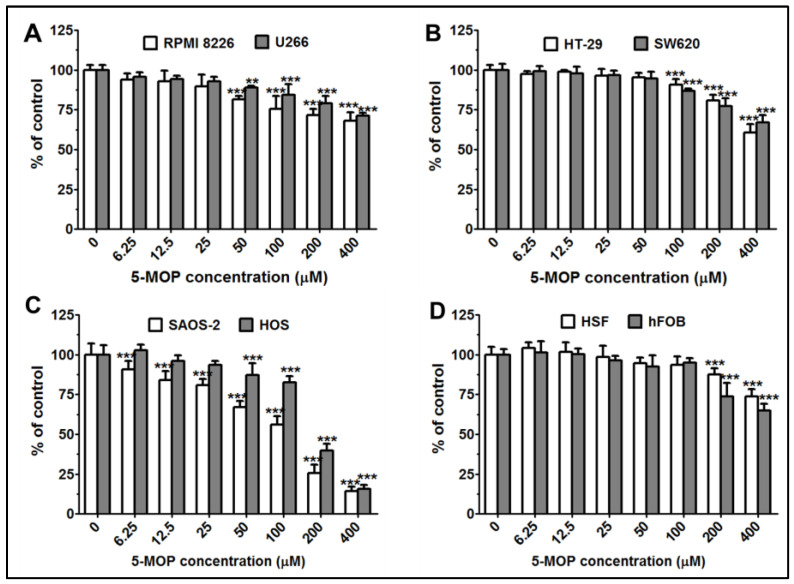
Effect of 5-MOP treatment on the viability of cancer and non-cancerous cells. (**A**) Multiple myeloma cells: RPMI 8226 and U266; (**B**) colon cancer cells: HT-29 and SW620; (**C**) osteosarcoma cells: Saos-2 and HOS; (**D**) normal cells: HSF and hFOB. The cells were cultured alone or in the presence of increasing concentrations of the compound, and cell viability was estimated using an NR assay at 48 h. Each bar represents the mean ± standard deviation (SD) of three independent experiments in comparison to the control (DMSO). One-way ANOVA test; ** *p* < 0.01 and *** *p* < 0.001.

**Figure 4 ijms-24-15555-f004:**
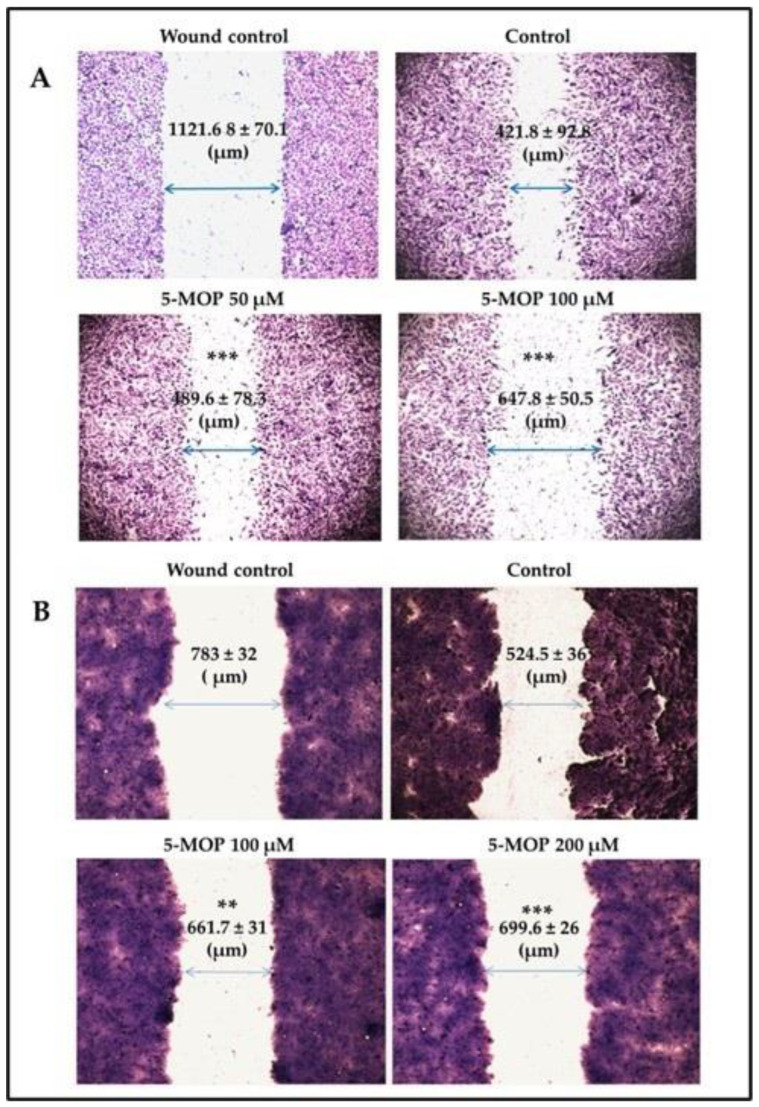
Effect of 5-MOP treatment on the migration of cancer cells. Wounded monolayers of cells were cultured alone or in the presence of the compound (50–200 µM) for 24 h. A representative micrograph of wound healing assay is shown for (**A**) Saos-2 cells and (**B**) HT-29 cells (magnification of 40×). 5-MOP at the indicated concentrations modulated the migratory phenotype of malignant cells in comparison to the control (DMSO). Statistically significant at ** *p* < 0.01 and *** *p* < 0.001 in a one-way ANOVA test.

**Figure 5 ijms-24-15555-f005:**
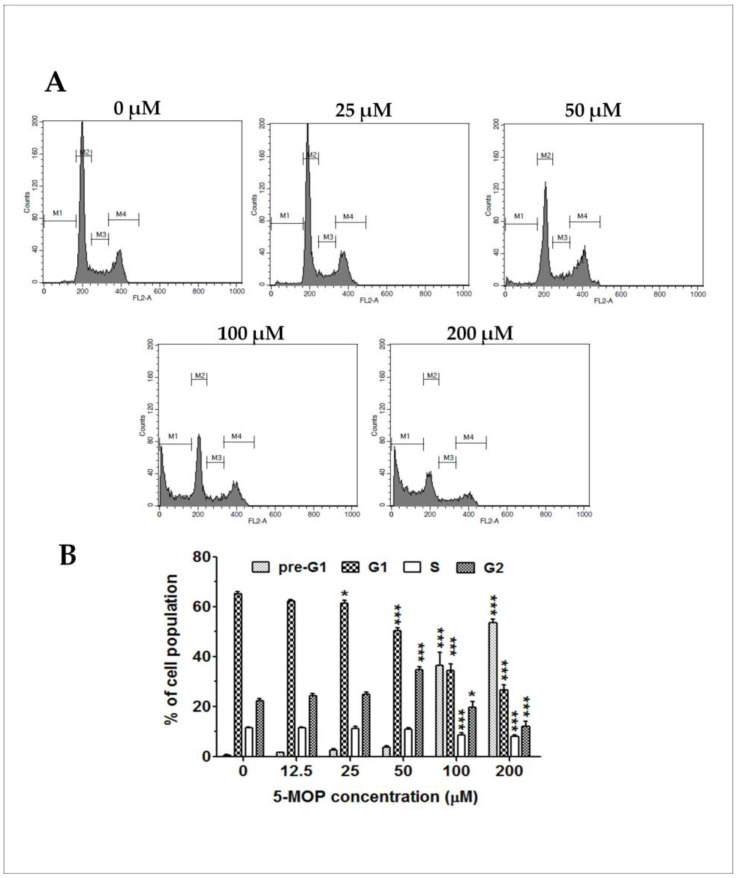
5-MOP treatment induced changes in the cell cycle profile of the Saos-2 cell line. The cells were cultured alone or in the presence of increasing concentrations of the compound. The cell cycle distribution in PI-labeled cells was analyzed via flow cytometry at 48 h. (**A**) The representative DNA histograms of the studied cells; (**B**) statistical analysis of the percentages of cells in the sub-G1, -G1, -S, and -G2 phases. Each bar represents the mean ± standard deviation (SD) of three independent experiments in comparison to the control (DMSO). * *p* < 0.05 and *** *p* < 0.001, one-way ANOVA test.

**Figure 6 ijms-24-15555-f006:**
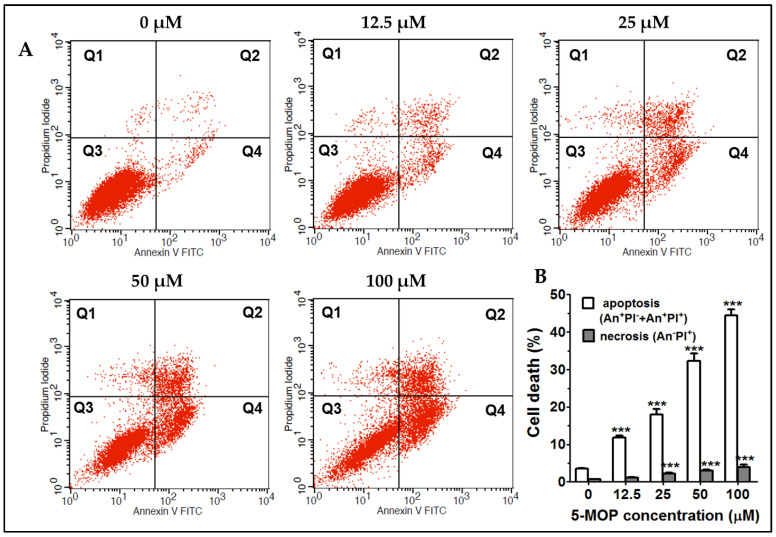
Effect of 5-MOP treatment on apoptotic cell death. (**A**) Flow cytometry analysis of apoptosis and necrosis in Saos-2 cell cultures after 48-h exposure to increasing concentrations of the compound. The apoptotic cell populations in quadrants Q2 and Q4 represent early and late apoptosis, respectively. The Q1 and Q3 quadrants represent necrotic and normal cell populations, respectively. (**B**) Statistical analysis of the percentages of apoptotic and necrotic cells. Each bar represents the mean ± standard deviation (SD) of three independent experiments in comparison to the control (DMSO). *** *p* < 0.001, one-way ANOVA test.

**Figure 7 ijms-24-15555-f007:**
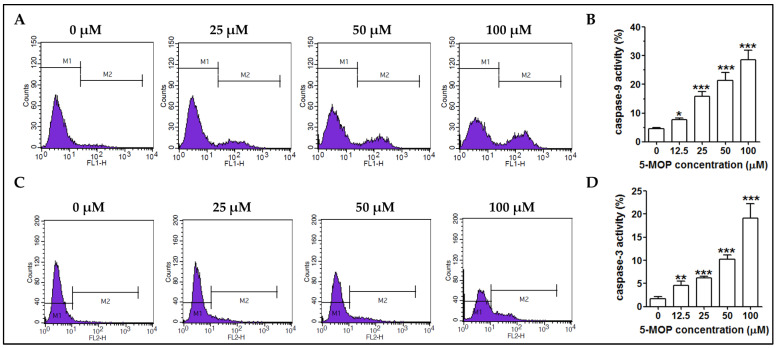
Caspase-mediated cell death is involved in the 5-MOP killing effect. Representative flow cytometry data from caspase-9 (**A**) and caspase-3 (**B**) assays with Saos-2 cells upon 48 h of exposure to increasing concentrations of the compound. M1 and M2 signified viable and apoptotic (caspase-positive) cells, respectively. The percentage of activated/cleaved caspase-9 (**C**) and caspase-3 (**D**) positive cells. Each bar represents the mean ± standard deviation (SD) of three independent experiments in comparison to the control (DMSO). * *p* < 0.05, ** *p* < 0.01, and *** *p* < 0.001, one-way ANOVA test.

**Figure 8 ijms-24-15555-f008:**
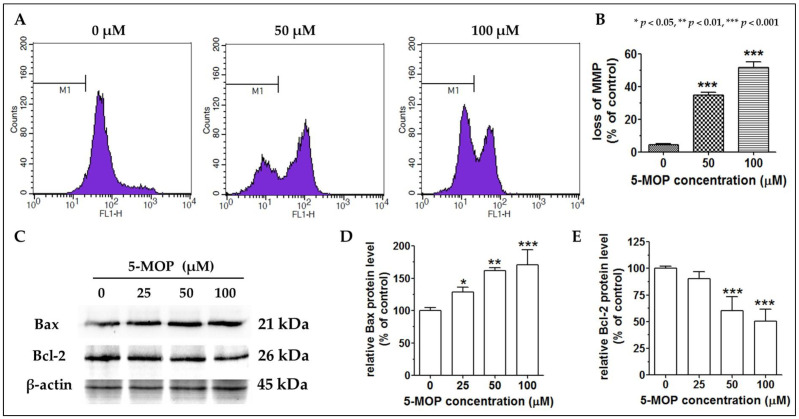
Effect of 5-MOP treatment on mitochondrial membrane potential (MMP) and expression of apoptosis-related proteins in Saos-2 cells. MMP was quantified after 48-h treatment via staining the cells with DiOC6, as reported in the Methods section. (**A**) Representative flow cytometry histograms for MMP measurements (the lower fluorescence values imply a lower MMP; M1 signified DiOC6 unstained cells). (**B**) Statistical analysis of the percentage of non-fluorescent cells. Each bar represents the mean ± standard deviation (SD) of three independent experiments in comparison to the control (DMSO). *** *p* < 0.001, one-way ANOVA test. The key markers of apoptosis, such as Bax and Bcl-2, were measured at 48 h via Western blotting. The Β-actin expression level served as a loading control. (**C**) Representative blots of whole-cell lysates. (**D**,**E**) Quantitative analysis of band intensities (Bax and Bcl-2, respectively) with Image J. v. 1.53t. (LOCI, Madison, WI, USA). Mean ± SD of three independent experiments in comparison to the control (DMSO). * *p* < 0.05, ** *p* < 0.01, and *** *p* < 0.001, one-way ANOVA test.

**Figure 9 ijms-24-15555-f009:**
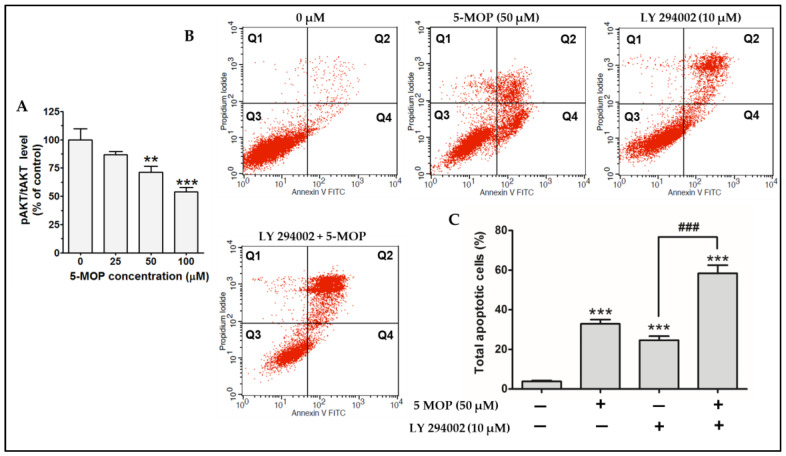
Effect of 5-MOP treatment on the phosphorylation of AKT kinase and the influence of the selective PI3K inhibitor (LY294002) on 5-MOP-triggered apoptosis in Saos-2 cells. The cells were incubated without or with the compound for 24 h and phosphorylated, and the total AKT was detected with an ELISA assay. (**A**) Quantification of the amounts of phosphorylated AKT^308^ compared to the total AKT kinase (pAKT/tAKT). The cells were treated with 5-MOP (50 µM), LY294002 (10 µM), or 5-MOP (50 µM) combined with LY294002 (10 µM) for 48 h, and the apoptosis rates were determined using a flow analyzer. (**B**) The representative dot plots indicate the percentage of An^−^/PI^+^ necrotic cells (Q1), An^+^/PI^+^ late apoptotic cells (Q2), An^−^/PI^−^ viable cells (Q3), and An^+^/PI^−^ early apoptotic cells (Q4) in the 5-MOP or/and LY294002-treated Saos-2 cell cultures. (**C**) Quantitative percentage of apoptotic (early + late apoptosis) cells in the control, 5-MOP, LY294002, and (LY294002 + 5-MOP)-treated Saos-2 cell cultures. Data are expressed as means ± SDs for three independent experiments. ** *p* < 0.01 and *** *p* < 0.001 in comparison to the control (DMSO); ### *p* < 0.001 in comparison to the LY294002-treated-cells; one-way ANOVA test.

## Data Availability

Not applicable.

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
