# Peer review of "Evaluation of the Biological Effect of Non-UV-Activated Bergapten on Selected Human Tumor Cells and the Insight into the Molecular Mechanism of Its Action"

_ijms, 2023, doi:10.3390/ijms242115555_

Round 1

Reviewer 1 Report

Bergapten, a naturally occurring furocoumarin, has gained increasing attention for its medicinal properties. In this manuscript, 5-methoxypsoralen (5-MOP) extracted from fruits was subjected to biological evaluation across various human cancer cell lines and healthy cell lines. The results of this in vitro investigation revealed a wide range of concentration-dependent cytotoxic effects against cancer cells associated with 5-MOP, while demonstrating the potential for minimal toxicity in normal cells. These findings suggest that 5-MOP holds promise as a prospective substance of natural origin for cancer treatment.

Overall, this manuscript is well structured and clearly written. Following are suggestions to strengthen the impact of the manuscript:

Major:

1.     None

Minor:

1.     Line 103: Please provide LC-MS and NMR spectrum of purified bergapten used in this study.

2.     Line 171: typo? “n case of”

Reviewer 2 Report

Dear authors,

Thank you for submitting your work. In your manuscript, you evaluated the anti-cancer effect of bergapten on several selected human tumor cells and further investigated the molecular mechanism behind it. The paper is well-organized, easy to follow, and provides a thorough discussion of the findings. It can be improved in the following aspects.

  1. Typos, missing commas, or other grammar issues (not limited to)
    1. line 27, line 37, line 45, line 49, line 60, line 62-64, line 68, line 70, line 77, line 99, line 105 (CPE not explained), line 109, line 171, line 234 (OS cells?), line 352
    2. IC50 and IC50 were used inconsistently
  2. Why adopting one-way ANOVA for the statistical analysis when you were comparing the means of two groups? T-test should be used in such scenarios whereas ANOVA is preferred for comparing three or more groups.
  3. In Figure c2, why HOS cells showed increased proliferation with 5-MOP concentration ranged from 6.25 to 100?
  4. Figure 6 was not aligned to the center.

See above.

Reviewer 3 Report

Magdalena Bartnik et al. in” Evaluation of the biological effect of non-UV-activated bergapten on selected human tumour cells and the insight in the molecular mechanism of its action.” show the anti-tumour potential of non-UV-activated 5-MOP in various human cancer cell lines, including HT-29 and SW620 (colorectal adenocarcinoma), U266 and RPMI8226 (multiple myeloma), Saos-2 and HOS (osteosarcoma). Impotantly, these lines, except one (HT29), were not previously tested in assessment of bergapten toxicity.

The article is well written and original.

The authors should explain the differences and possible similarities between the different cell lines used in the study.

Minor editing of English language required

Reviewer 4 Report

In this manuscript, " Evaluation of the biological effect of non-UV-activated bergapten on selected human tumour cells and the insight in the molecular mechanism of its action, " the author explored the antitumor role of 5-MOP on Osteosarcoma proliferation, invasion and tried to explore the molecular mechanisms. I have the following queries/suggestions.

1. Figure 1A: The graph should include the concentration of 5-MOP at its IC50 dose. Utilizing a line graph with a logarithmic scale to display all doses would enhance clarity.

2. It's essential for the author to explain the protumor effect of 5-MOP in HOS cells at concentrations below 100 µM.

3. As observed, 5-MOP leads to cytotoxicity in non-cancerous cells HFS and hFOB at concentrations above 200 µM. This raises questions about whether the effect on tumor cells could be generalized cytotoxicity. Hence, concluding that tumor cells have an IC50 value of more than 200 µM may be inaccurate.

4. Specify the concentration of 5-MOP used in the wound healing experiment shown in Figure 4.

 5. The legend for Figure 7 needs correction in labeling. The significance of caspase expression is only in the cleaved and active form; otherwise, it is inactive. The author showed the Caspase 3&9. Is it total or cleaved caspase level?? Should include the cleaved and total caspase level in the study.

6. The discussion around Akt activation should specify that it primarily occurs through phosphorylation at Serine 473 and Threonine 308. Analyzing relative phosphoprotein expression at these sites and comparing it to the same total protein is more reliable for phosphoprotein analysis. The analysis of total Akt phosphorylation alone may be misleading, and conclusions based on it may be unreliable.

7. In Figure 9, the combination experiment appears to show an additive effect of 5-MOP and LY294002 on tumor cell viability. Hence, it is unclear how the author concluded that 5-MOP is involved in the Akt pathway in its cytotoxic effects.

8. The discussion section is overly detailed, focusing too much on the background of molecules involved, and lacks coherence. The author should emphasize critically analyzing and interpreting the findings and relating them to the context of the published literature. Extensive rewriting is necessary.

9. The manuscript should provide a complete form of each abbreviation as they appear for the first time to enhance reader understanding.

Round 2

Reviewer 4 Report

None